

# Human agency beliefs influence behaviour during virtual social interactions

Nathan Caruana[1,2,3,4,*], Dean Spirou[5,*] and Jon Brock[3,4,5]

[1] Department of Cognitive Science, Macquarie University, Sydney, New South Wales, Australia
[2] Perception in Action Research Centre, Sydney, New South Wales, Australia
[3] Centre for Atypical Neurodevelopment, Sydney, New South Wales, Australia
[4] ARC Centre of Excellence in Cognition and its Disorders, Syndey, New South Wales, Australia
[5] Department of Psychology, Macquarie University, Sydney, New South Wales, Australia
[*] These authors contributed equally to this work.

Corresponding author
Nathan Caruana,
nathan.caruana@mq.edu.au

## ABSTRACT

In recent years, with the emergence of relatively inexpensive and accessible virtual reality technologies, it is now possible to deliver compelling and realistic simulations of human-to-human interaction. Neuroimaging studies have shown that, when participants believe they are interacting via a virtual interface with another human agent, they show different patterns of brain activity compared to when they know that their virtual partner is computer-controlled. The suggestion is that users adopt an "intentional stance" by attributing mental states to their virtual partner. However, it remains unclear how beliefs in the agency of a virtual partner influence participants' behaviour and subjective experience of the interaction. We investigated this issue in the context of a cooperative "joint attention" game in which participants interacted via an eye tracker with a virtual onscreen partner, directing each other's eye gaze to different screen locations. Half of the participants were correctly informed that their partner was controlled by a computer algorithm ("Computer" condition). The other half were misled into believing that the virtual character was controlled by a second participant in another room ("Human" condition). Those in the "Human" condition were slower to make eye contact with their partner and more likely to try and guide their partner before they had established mutual eye contact than participants in the "Computer" condition. They also responded more rapidly when their partner was guiding them, although the same effect was also found for a control condition in which they responded to an arrow cue. Results confirm the influence of human agency beliefs on behaviour in this virtual social interaction context. They further suggest that researchers and developers attempting to simulate social interactions should consider the impact of agency beliefs on user experience in other social contexts, and their effect on the achievement of the application's goals.

## INTRODUCTION

The development in recent years of relatively inexpensive and accessible virtual reality technology now makes it possible to deliver compelling and realistic simulations of human-to-human interaction (*Georgescu et al., 2014*; *Schroeder, 2002*). Potential applications of virtual social interaction are only starting to be explored, but already include gaming, market research, basic and clinical scientific research, military simulation training, long distance health care delivery and education (*Lee & Stewart, 2016*). Some of these applications involve the co-presence of two or more human-controlled avatars within the same virtual environment. However, when the interaction is to be delivered to a large number of users (e.g., for standardised education and training delivery) or when tight control of the interaction is required (e.g., in social cognition and neuroscience research), it may be possible and desirable for individual humans to interact with a virtual character whose behaviour is entirely controlled by a computer algorithm (*Caruana et al., 2017a*; *Georgescu et al., 2014*). In such contexts, an important question is the extent to which the user experience is affected by their knowledge that the social interaction is artificial. In other words, does the user interact differently with a virtual social partner if they know that the partner is a computer-controlled agent rather than a human-controlled avatar?

Central to this question is the observation that humans negotiate everyday social interactions by *mentalising*—interpreting the behaviour of social partners in terms of mental states such as beliefs, desires, and goals—and then using those inferred mental states to predict future behaviour and adapt their responsive behaviour accordingly (*Premack & Woodruff, 1978*). To give a concrete example, suppose you are walking towards someone on a crowded footpath, and your eyes meet. At once, you are mutually aware of each other and can make a joint effort to avoid bumping into one another. However, if you see that the other person is looking at the displays in the shop windows as they walk down the path, you will predict that they will continue to walk towards you unaware, and must change your own trajectory to avoid a collision. If we believe that a virtual character is controlled by another human then we are likely to engage in these same mentalising processes, adopting what philosophers refer to as an "intentional stance", because we see the agent's behaviour as a product of an intentional and intelligent "mind" (*Wykowska et al., 2014*). The question is, what happens when we know or believe that the virtual character is artificial? Does our interpretation of its behaviour—and therefore our response—change? Or does a sufficiently realistic virtual partner elicit the adoption of an intentional stance even when we consciously know that our partner lacks human agency and, therefore, mental states?

In the current study, we addressed these questions in the context of a gaze-based "joint attention" interaction—in which participants interacted with a virtual partner to reach a common focus of attention. In a typical joint attention episode, one person initiates joint attention (IJA) by directing their partner's attention to an object or location in space (*Bruinsma, Koegel & Koegel, 2004*). The second person responds to the joint attention bid (RJA), and, finally, the first person monitors the behaviour of the second to determine whether joint attention has been achieved (*Bruinsma, Koegel & Koegel, 2004*).

Joint attention is reciprocal, dynamic, and intentional (*Schilbach et al., 2013*). It also requires individuals to represent the mental states of others (e.g., What is my partner looking at? Are they attempting to communicate with me? etc.). Thus, joint attention provides a useful model of social interaction for investigating the effects of agency beliefs during virtual interactions.

Gaze-based joint attention is a particularly interesting avenue of research, given that eyes—unlike other human sensory organs—have the capacity to both perceive and display communicative signals. Recent work has revealed that social context (e.g., believing whether one's own eye movements will be seen by another person in a two-way interaction) can influence communicative eye movement behaviour (*Gobel, Kim & Richardson, 2015*). Until now, however, the gaze processing literature has largely been restricted to the investigation of gaze *perception* in non-interactive contexts (see *Schilbach et al., 2013* for a review). This highlights the need to investigate the influence of social context on interactive gaze behaviour where eye gaze is simultaneously used as a cue to understand others (perception) as well as a communication mechanism (signaling; *Gobel, Kim & Richardson, 2015*). By using interactive tasks which measure joint attention behaviour, we are able to investigate the influence that human agency beliefs have on the way we perceive and respond to gaze (RJA) and use our own gaze to communicate (IJA).

Our joint attention task builds on several recent neuroimaging studies of joint attention (see *Caruana et al., 2017a*) in which participants' eye-movements are tracked as they interact with an animated virtual character, whose own eye-movements are responsive to those of the participant. In some studies, participants have been told that the "avatar" is controlled by a second participant whose eye movements are also being recorded (e.g., *Schilbach et al., 2010*). In other studies, participants know that their partner is computer-controlled (e.g., *Oberwelland et al., 2016*). Recently, two studies have directly investigated whether these different approaches, and the adoption of human agency beliefs, influence brain activity during joint attention experiences. In a study by Pfeiffer and colleagues (*2014*) participants initiated joint attention bids towards a target, and their virtual partner responded by either looking towards or away from the target. For each block of trials, participants were required to indicate whether they believed their virtual partner was being controlled by a human or computer. Although, in reality, the virtual character was always computer-controlled, blocks of trials in which participants believed they were interacting with another human were associated with increased activation of the ventral striatum—a brain region associated with reward processing. However, in this study, agency beliefs were confounded with task success (i.e., achieving joint attention), as participants in the naïve condition were more likely to say that the avatar was human-controlled on blocks when he was more responsive. Thus, striatal activity may simply reflect task success irrespective of agency beliefs.

In a second study measuring event-related potentials (ERPs), we employed a similar task and a between-subjects design, informing half of the participants that the virtual character was human-controlled and half that he was computer-controlled (*Caruana, De Lissa & McArthur, 2017*). We found that the N170—an early occipitotemporal brain response to visual information—was larger in response to gaze shifts in the group who

believed the virtual character was human-controlled (see *Wykowska et al., 2014* for similar findings). We also found that the P350—a later response measured over centro-parietal sites—was sensitive to joint attention success only in the group who believed that the virtual character was human-controlled. As with the study by Pfeiffer and colleagues, the differential brain response suggests that participants process the outcome of a joint attention episode differently depending on their beliefs in the agency of their partner. However, it is important to note that, in both cases, the effect is driven by the behaviour of the virtual partner—whether he is programmed to respond correctly or not on each trial. These studies do not address the impact of agency beliefs on participants' own behaviour during the interaction; that is, how they respond to, and initiate joint attention bids.

In the current study, therefore, we investigated whether human agency beliefs have a direct influence on joint attention *behaviour*. As in the studies reviewed above, participants interacted with a virtual partner in a cooperative joint attention game. Half of the participants believed that their partner was controlled by another human (Human condition). The remainder were correctly informed that their partner was computer-controlled (Computer condition). Their task was to catch a burglar located in one of six houses placed around the edge of the screen (see Fig. 1). At the start of each trial, the participant and their partner searched their allotted houses and whomever found the burglar was then required to look back at the burglar to signal its location. The burglar was caught when both players were looking at the correct location. Unlike previous joint attention studies investigating the influence of human agency beliefs, this task created a context in which sometimes the participant found the burglar and had to "Initiate" joint attention, and other trials where they did not find the burglar, and had to "Respond" to their partner instead. In addition to this "Social" task, participants also completed a non-social "Control" task in which the virtual character's eyes remained closed and participants completed the same sequence of eye-movements in response to geometric shape cues (circles and arrows).

We have used this task in previous studies but without the agency manipulation. In other words, all participants believed that they were interacting with a real person (*Caruana, Brock & Woolgar, 2015*; *Caruana et al., 2017c*; *Caruana et al., 2017b*). These studies have produced a number of consistent findings which motivated our predictions in the current study.

First, on responding (RJA) trials, participants are slower to respond to their partner's eye gaze cue than for the equivalent arrow cue in the control (RJAc) condition. Importantly, this effect is reduced when the search phase is removed from the task so that the virtual partner only makes a single eye-movement on each trial (*Caruana et al., 2017b*). This suggests that an important part of the joint attention task is determining whether a shift in eye gaze is intended to be communicative or not. If participants know that their partner is not human and, therefore, has no mental states or intentions, they may not evaluate the communicative intent of their partner's behaviour in the same way. We therefore predicted that this effect would be reduced in the Computer condition compared to the Human condition.

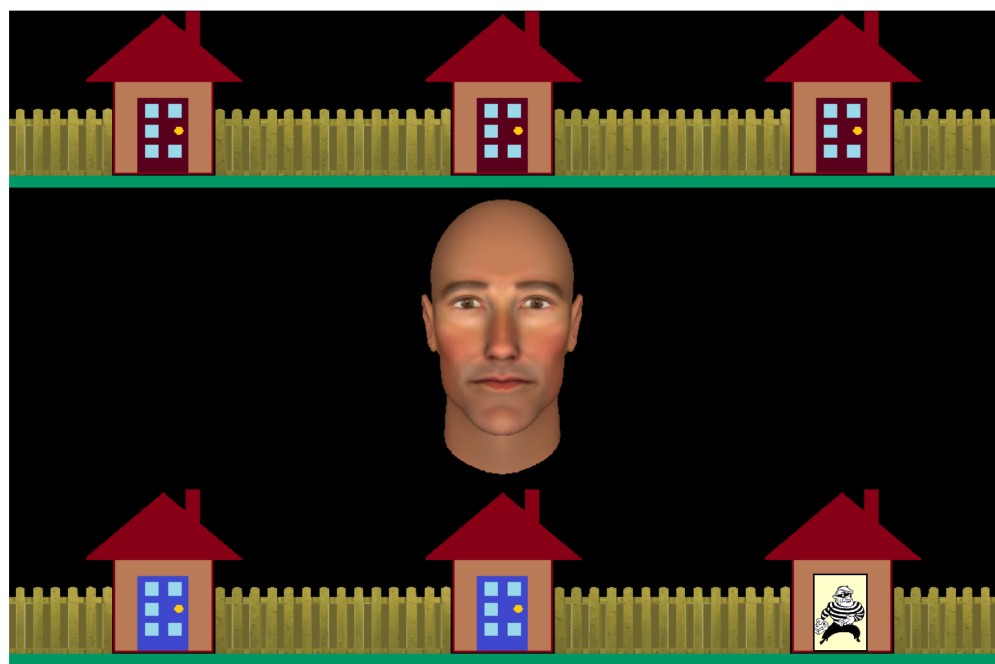

**Figure 1  A screenshot of the interactive task stimuli.** The row of houses with blue doors are to be searched by the participant to find the burglar. The houses with red doors are to be searched by the virtual partner depicted by the virtual character in the centre of the screen.

Second, on "Initiate" trials, participants discover the burglar and are then required to look back towards the avatar. We find they are slower to do this in the Social (IJA) condition than the Nonsocial (IJAc) control condition. They are then required to either wait for eye contact from their partner (IJA) or wait for the central fixation point to turn green (IJAc) before saccading back to the burglar location. We have found that participants make more premature saccades (i.e., failing to wait for the respective cue before looking back at the burglar) in the IJA condition compared to IJAc. Again, these findings can be interpreted in terms of the inferred mental states of the virtual partner. When participants think their partner is human, they assume that he will intuitively know that they are looking at a location to initiate joint attention, even when eye contact is not first established to signal their own communicative intent. In the Control condition, they know they are interacting with the computer and so approach the task quite differently, making the same robotic pattern of eye movements on each trial. Our prediction, therefore, is that both of these effects will be reduced when participants know that their virtual partner is computer-controlled. That is, they will approach the interaction with the virtual partner in a similar fashion to their "interaction" with the symbols on the screen. If these predictions were confirmed, they would provide the first direct evidence that beliefs about the human agency of virtual characters can influence user behaviour during virtual interactions.

## METHODS

### Ethical statement

This study was approved by the Macquarie University Human Research Ethics Committee (Approval reference: 5201200021). All participants provided verbal and written consent before participating in the study.

### Participants

Participants were first year undergraduate students at Macquarie University who received course credit for their involvement. They were alternately allocated to either the "human" or "computer" group in the order of participation. At the end of the experiment, two participants in the human group indicated that they did not believe that a human was controlling the virtual character. These two participants were excluded and replaced by the next participant in the testing schedule. The final sample included 48 participants, 24 in each group. The two groups were similar in terms of sex ratio (19 females in human group, 17 in the computer group) and age (Human: $M = 19.33$, $SD = 0.52$; Computer: $M = 19.51$, $SD = 0.35$). All participants had normal vision and reported having no psychiatric diagnoses or history of neurological impairment or injury. All participants were right-handed, as confirmed using the Edinburgh Handedness Inventory (*Oldfield, 1971*).

### Joint attention task

At the beginning of each session all participants completed the Oldfield Handedness Inventory. During this time, the experimenter (DS) told participants in the human condition that he was going to "check-up" on a colleague who would be assisting with the study, briefly left the room, entered the adjacent room for a minute, and then returned. The experimenter then read the same set of scripted instructions to participants in both groups using graphical cue cards (see Supplemental Information 1, cards 1, 7–12). Participants in the human group were then told that the experimenter's colleague, Alan, who was in the adjacent eye-tracking laboratory, would be controlling the avatar that they would be completing the task with. Participants were also told that their eye movements would be displayed on a virtual character that Alan could see on his screen. This deception was supported by additional instruction cards (Supplemental Information 1, cards 2–6) which explained how the interactive interface worked and illustrated Alan's view of the stimulus. Participants in the computer group were simply told (truthfully) that the avatar was controlled by a computer program.

The joint attention task was programmed using *Experiment Builder* 1.10.165 software (*SR Research, 2004*). It was identical for both groups (human and computer) and was also identical to that used in our previous studies (*Caruana, Brock & Woolgar, 2015*; *Caruana et al., 2017c*; *Caruana et al., 2017b*). Full details of the gaze-contingent algorithm can be found in *Caruana, Brock & Woolgar (2015)*.

The display comprised an anthropomorphic virtual character in the center of the screen subtending 6.5 degrees of visual angle (see Supplemental Information 2 for images of all avatar stimuli), with two horizontal rows of three houses, each subtending four degrees of visual angle, positioned above and below the virtual character (see Fig. 1). At the beginning

of each trial, participants were required to search the houses with a blue door by fixating them in turn, whereupon the doors open to reveal either an empty house or the burglar. The location of the blue doors (i.e., top versus bottom row of houses) changed from the first to the second block, and block order was counterbalanced across participants within each group. On some trials, one or two houses were already open to vary the participants' search behaviour across trials. Participants could search these houses in any order they chose.

## Social conditions (RJA and IJA)

Once the participant completed their search (either by finding the burglar or by discovering that all the blue houses were empty), they were required to look back at their partner to establish eye contact. The virtual character was programmed to search the red-doored houses in a random order until the participant had looked back at them and then to make 0–2 additional gaze shifts before establishing eye contact. The onset latency of each gaze shift varied between 500–1,000 ms. This meant that the delay between fixating the avatar's face and the establishment of eye contact varied between 500–3,000 ms.

### Responding to joint attention (RJA)

In RJA trials, the participant would find all the blue-doored houses empty, indicating that the burglar was in one of the red-doored houses. Once eye contact had been established, the virtual character would look towards the red door concealing the burglar. If the participant responded by looking at that door, it would open to reveal the burglar behind bars to indicate that he had been captured.

### Initiating joint attention (IJA)

In IJA trials, the participant would find the burglar in one of the blue-doored houses. Following eye contact, the participant was required to conduct an "initiating saccade" to the location of the burglar by fixating back on the house that contained the burglar. At this point, the virtual character would follow the participant's gaze. If this was the correct location, then the burglar again reappeared behind bars. Importantly, the virtual character did not respond if the participant made their initiating saccade prior to the establishment of eye contact (this was classified as a premature saccade). However, the trial could still be completed if the participant looked back at their partner, established eye contact, and made a second initiating saccade back to the burglar.

### Feedback

On correct trials, participants were informed that they had successfully caught the burglar if the burglar appeared behind bars. On incorrect trials, participants were presented with the burglar in red at the correct location to indicate that they were unsuccessful in catching the burglar. An incorrect trial could be the result of a 'location error' or a 'timeout error'. A location error occurred when participants fixated the wrong location when responding to or initiating a joint attention bid. A timeout error occurred when participants failed to respond to or initiate a joint attention bid within three seconds of being guided on RJA trials, and establishing eye contact on IJA trials respectively. Finally, a Search Error occurred, and participants were presented with a "Failed Search" error message if they spent

more than three seconds fixating away from their designated houses before completing their search for the burglar. If this occurred, the trial was terminated and removed from all analyses.

### Non-social conditions (RJAc and IJAc)

To control for the non-social task requirements in both the RJA and IJA task conditions (e.g., task complexity, attention and action inhibition), two non-social conditions were included. In these conditions, participants completed the same task without any social interaction. The virtual character stimulus remained on the screen, however the eyes were closed for the duration of the trial. A grey fixation point was placed in the center of the animated face, which participants were required to fixate once completing their search for the burglar. This turned green after 500–3,000 ms (analogous to establishing eye contact). On RJAc trials this was followed by the presentation of a green arrow, which indicated the burglar's location (analogous to the virtual partner's guiding eye gaze), which participants were to follow. On IJAc trials, participants were required to fixate back on the burglar's location to catch the burglar once the fixation point turned green.

### Procedure

Participants completed two blocks, each comprising 108 trials. This included 27 trials per condition (i.e., IJA, RJA, IJAc RJAc). Within each block, Social (IJA, RJA) and Control (IJAc, RJAc) trials, were completed in clusters of six trials each. Whether each block began with a Social or Control cluster of trials was counterbalanced across subjects and matched between groups. The start of a Social cluster was cued with text reading "Together" and the start of a Control cluster was cued with text reading "Alone". These cues were presented in the centre of the computer screen for 1,000 ms each time.

### Eye-tracking

An EyeLink 1,000 Remote Eye-Tracking System (SR Research Ltd., Ontario, Canada) was used to track the participants' right eye movements with a sampling rate of 500 Hz, and a chin rest to stabilise head movements and standardise viewing distance. A 9-point eye-tracking calibration and validation was conducted at the beginning of each block.

### Subjective experience ratings and debrief

Following the completion of the joint attention task, participants completed a post-experimental interview where they were asked to rate how difficult, natural, intuitive and pleasant they found the Social and Control tasks on a 10-point Likert scale (1 = *Not at all,* 10 = *Extremely*). Participants also rated how co-operative their partner was on Social trials, and how "human-like" the virtual character felt generally, as well as how human-like he appeared and behaved, using the same 10-point scale. Participants were also asked whether they preferred completing the task alone (Control trials) or together with their partner (Social trials), and rated the strength of this preference on a 10-point scale (1 = *completely prefer together,* 10 = *completely prefer alone*). Participants in both the Human and Computer group were asked the same questions. The questions were designed to gauge the extent to which participants anthropomorphised the virtual character during

their interaction with him, and to provide participants in the Human group with the opportunity to disclose whether they had any doubts that they were truly interacting with another human being.

At the end of the session, participants in the Human group were debriefed about the true nature of the interaction. At this point they were asked whether they believed they were interacting with another person named Alan. Participants also rated how convinced had been on the same 10-point scale described above.

## Analysis

We used DataViewer software (SR Research Ltd., Ontario, Canada) to export Interest Area and Trial reports. All subsequent analyses were performed in R using a custom script. Raw data and R code can be downloaded from the Open Science Framework: https://osf.io/yqb7g/. R Markdown can also be viewed here: http://rpubs.com/JonBrock/Belief.

Following our previous studies (*Caruana, Brock & Woolgar, 2015*; *Caruana et al., 2017c*; *Caruana et al., 2017b*), we excluded all trials in which a recalibration was required or an error occurred during the search phase. We then measured the following indices of performance:

### Accuracy

Proportion of trials on which the participant successfully caught the burglar.

### Saccadic reaction time (Respond trials)

Mean duration between the presentation of the gaze (RJA) or arrow (RJAc) cue and the onset of the participant's responding saccade to the burglar location. Trials with incorrect responses or reaction times below 150 ms were excluded. The trial timed out at 3,000 ms (and was coded as an error).

### Target dwell time (Initiate trials)

Mean duration between the burglar being revealed and the participant looking back towards Alan (IJA) or the fixation point (IJAc). Trials with dwell times below 150 ms or above 3,000 ms were excluded.

### Premature initiating saccades (Initiate trials)

Proportion of trials in which a saccade was made from the avatar (IJA) or fixation point (IJAc) to the location of the burglar, prior to the establishment of eye contact (IJA) or the grey fixation point turning green (IJAc).

## Statistical analysis

Statistical analyses were conducted using the ez package (version 4.4-0; *Lawrence, 2016*) in R. We conducted mixed-ANOVA with condition (i.e., Social versus Control) as a within-subjects factor and group (i.e., Human versus Computer) as a between-subjects factor. We have reported generalised eta squared ($\eta_G^2$) as a measure of effect size. Interactions were followed up with $t$-tests. For the subjective ratings, we used non-parametric Mann–Whitney $U$ tests to investigate the effect of group for each rating. For Accuracy analyses, we also examined the additional factor of subject role (i.e., Responding vs Initiating).

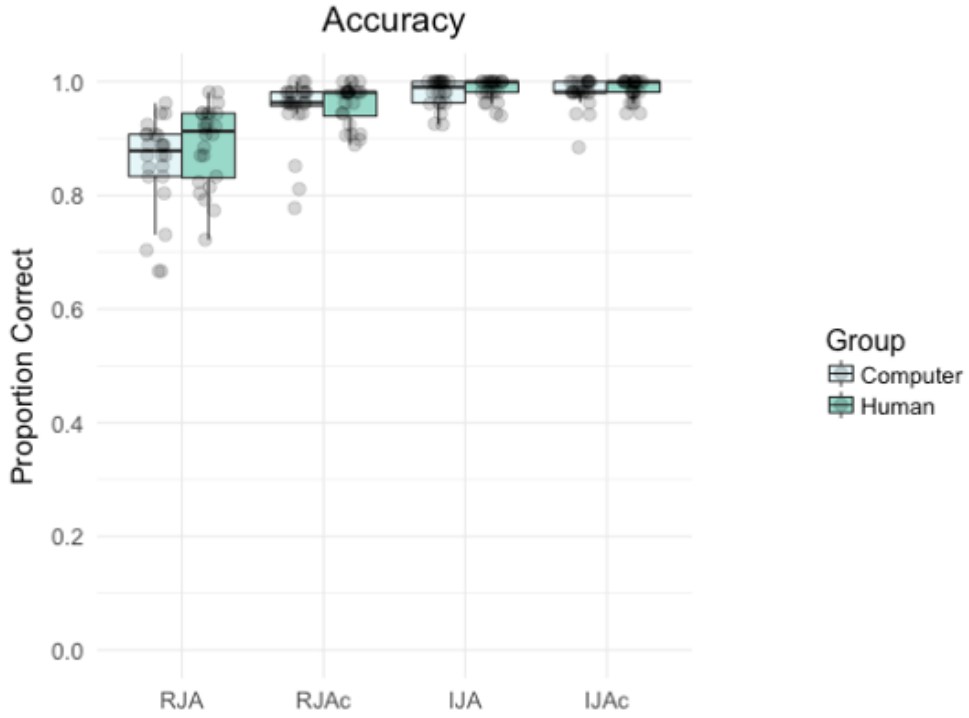

**Figure 2** **Boxplot with individual data points depicting the proportion of correct trials by group (i.e., Computer, Human) and condition (i.e., RJA, RJAc, IJA, IJAc).** In all figures, whiskers extend (as in a conventional Tukey boxplot) to the furthest data points that are within 1.5 times the length of the box from the end of the box.

## RESULTS

### Accuracy

Figure 2 summarizes accuracy by group and condition. Participants made significantly more errors on Social trials (RJA, IJA) than Control trials (RJAc, IJAc) overall, (main effect of condition, $F(1,46) = 43.86$, $p < .001$, $\eta_p^2 = 0.14$), and more errors on Responding trials (RJA, RJAc) than Initiating trials (IJA, IJAc) overall (main effect of social role, $F(1,46) = 95.87$, $p < .001$, $\eta_p^2 = 0.14$). There was also a significant group*condition*subject role interaction ($F(1,46) = 0.36.69$, $p < .001$, $\eta_p^2 = 0.15$) which arose because participants made more errors on RJA trials than any other condition. There were no significant main effects of group, nor any significant interactions involving group (all $ps > 0.08$, see https://osf.io/yqb7g/ for full analyses).

### Responding to joint attention
*Saccadic reaction time*

Figure 3A summarises saccadic reaction time data by group and condition. Participants' saccadic reaction times were significantly slower on RJA trials relative to RJAc trials (main effect of condition, $F(1,46) = 264.63$, $p < .001$, $\eta_G^2 = 0.66$). Overall, saccadic reaction times in the Computer group were significantly slower than the Human group (main effect of group, $F(1,46) = 5.71$, $p = .021$, $\eta_G^2 = 0.08$). However, there was no significant group*condition interaction ($F(1,46) = 2.34$, $p = .133$, $\eta_G^2 = .02$).
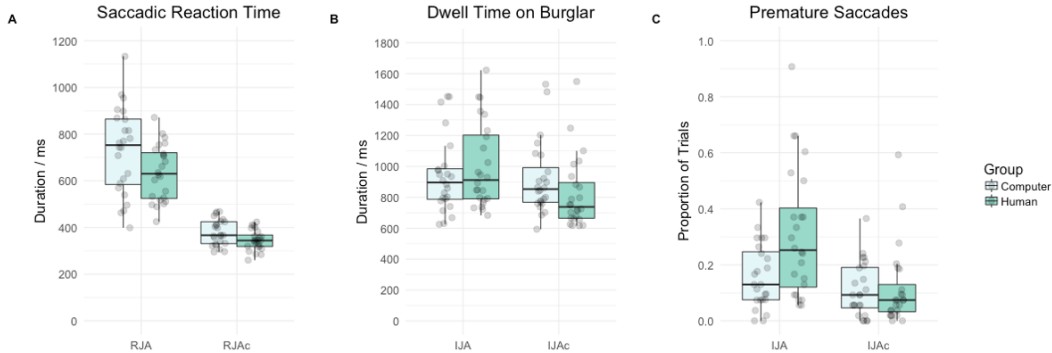

**Figure 3** Tukey boxplot with individual data points depicting (A) saccadic reaction times in milliseconds (B) dwell time on the burglar in milliseconds and (C) the proportion of trials participants made a premature saccade by group (i.e., Computer, Human) and condition (i.e., RJA versus RJAc for A, or IJA versus IJAc for B and C).

## Initiating joint attention

### Target dwell time

Figure 3B summarises target dwell time data by group and condition. There was no significant main effect of group ($F(1,46) = 0.06$, $p = .816$, $\eta_G^2 = 0.00$). However, as anticipated, participants had significantly longer dwell times for the burglar on IJA trials compared to IJAc trials (main effect of condition, $F(1,46) = 24.36$, $p < .001$, $\eta_G^2 = 0.04$). More importantly, in line with our hypotheses, there was also a significant group*condition interaction ($F(1,46) = 14.72$, $p < .001$, $\eta_G^2 = 0.03$). Follow-up $t$-tests revealed that participants had significantly longer dwell times on the burglar on IJA trials compared to IJAc trials, in the Human group ($t(23) = 5.58$, $p < .001$) but not in the Computer group ($t(23) = 0.89$, $p = .383$).

### Premature initiating saccades

Figure 3C summarises the proportion of successful trials in which participants made a premature initiating saccade, by group and condition. Again, there was no significant main effect of group ($F(1,46) = 3.78$, $p = .058$, $\eta_G^2 = 0.06$). However, as predicted, participants made significantly more premature initiating saccades on IJA trials than on IJAc trials (main effect of condition, $F(1,46) = 38.50$, $p < .001$, $\eta_G^2 = 0.14$). Of greater interest, and again aligning with our hypotheses, we found evidence of a significant group*condition interaction ($F(1,46) = 13.79$, $p < .001$, $\eta_G^2 = 0.06$). Follow-up $t$-tests revealed that participants made significantly more premature initiating saccades on IJA trials compared to IJAc trials in the Human group ($t(23) = 6.15$, $p < .001$) and the Computer group ($t(23) = 2.11$, $p = .046$)

## Subjective task ratings

Figure 4 provides a summary of the subjective task ratings involving the social condition. Participants in the Human group rated their partner as being significantly more cooperative compared to participants in the Computer group $W = 193.0$, $p = .039$. They also found the task more pleasant, $W = 188.5$, $p = .038$, but less intuitive, $W = 384.5$, $p = .045$. There were no significant differences between groups in any other ratings.

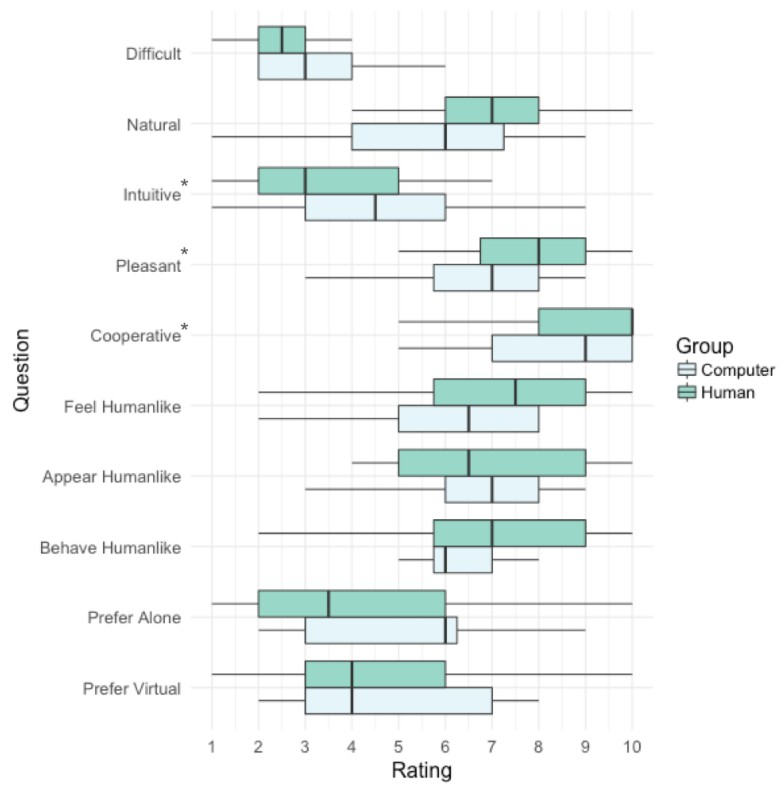

**Figure 4** Tukey boxplots depicting responses to subjective ratings questions: (1) *Difficult*, How difficult did you find the interactive task? (2) *Natural*, How natural did the interaction feel? (3) *Intuitive*, How intuitive was the interactive task? (4) *Pleasant*, How pleasant was the interaction? (5) *Cooperative*, How cooperative did you think your partner was? (6) *Feel Humanlike*, How human-like did the avatar feel? (7) *Appear Humanlike*, How human-like was the avatar's appearance? (8) *Behave Humanlike*, How human-like was the avatar's behaviour? (9) *Prefer Alone*, Which task did you prefer most? The interactive or the solo task? (10) *Prefer Virtual*, Would you prefer to play this game face-to-face or using virtual reality?

### Belief of the agency of the virtual character

In the Human group, all participants other than two excluded participants (see Participants section) reported that they were convinced they were interacting with another human being through the virtual interface.

## DISCUSSION

The current study investigated the effect of human agency beliefs on behaviour during virtual joint attention interactions. Although overall task accuracy was equivalent in our two groups, participants who believed that their virtual partner was controlled by a real person showed markedly different patterns of eye-movements and response times compared to participants who knew that their partner was computer-controlled. As discussed below, these findings indicate that human agency beliefs affect expectations of a virtual partner's behaviour, responsiveness and flexibility, as well as perhaps the human's own social motivation.

## Responding to joint attention

The results observed in the Human group replicated the findings of our previous studies in which all participants have believed their virtual partner to be human-controlled (*Caruana, Brock & Woolgar, 2015*; *Caruana et al., 2017c*; *Caruana et al., 2017b*). In the Respond conditions, participants responded more slowly to the eye gaze cue during the Social condition (RJA) than the arrow cue in the Control condition (RJAc). As noted earlier, this effect is partially attributable to the ambiguity of the eye gaze cues, which occur in the context of multiple non-communicative eye-movements made by the virtual character. This requires participants to engage in a process that we call "intention monitoring" (see *Caruana et al., 2017b*) because participants are required to infer the "communicative intent" of their partner's gaze cues. However, contrary to expectation, we did not find a reduction of the effect in the current Computer group. Despite knowing that their partner was not real and therefore had no intentions, participants were still significantly slower to respond to the eye gaze cue than the arrow cue, with the effect being similar in magnitude to the Human condition.

One explanation for this finding is that the nature of the RJA task uniquely encouraged participants to evaluate the avatar's behaviour in the same way they would an intentional agent. Even though participants in the Computer group did not believe that they were interacting with an intentional agent, they were still required to process the cues that usually signal intentional and communicative social behaviour (i.e., eye contact) in order to selectively respond to the gaze cues that signalled the burglar's location, and disambiguate these joint attention bids from earlier gaze shifts that were part of their partner's search for the burglar. This cue-disambiguation process was absent on the non-social RJAc trials, where participants responded to unambiguous arrow cues, which explains why we see a difference in saccadic response times on RJA and RJAc trials for both groups. The additional task demands placed on participants during RJA trials may also explain why accuracy was significantly lower for these trials compared to RJAc trials across both groups.

Although we did not find an interaction between Condition and Group for Responding trials, we did find a main effect of Group with participants in the Human group responding faster on Social *and* Control trials compared to participants in the Computer group. This may be because participants in the Human group were more motivated than those who believed that they were interacting with a computer. In other words, the perceived presence of a human co-operator produced an 'audience effect' or social pressure which induced faster responses across both social and non-social trials (*Park & Catrambone, 2007*). Indeed, this interpretation does align with some of the incidental comments made by some participants in the Human group of this study, and other studies that we have run in the past, in which they would say things like "*I didn't want to let the other guy down*" or "*I felt that Alan was better at the task than I was*".

## Initiating joint attention

Results from the Human group also replicated our previous findings for the Initiating joint attention condition. Firstly, we found no effects of condition (or group) for accuracy on Initiating trials. Consistent with our previous work (cf. *Caruana et al., 2017c*), ceiling

effects were observed across all participants and conditions. This is likely due to the fact that, upon finding the burglar, participants had an unlimited amount of time to complete IJA and IJAc trials. This allowed us to investigate the pattern of eye-movement behaviours reported in our subsequent analyses, but also made it difficult for participants to fail.

Compared to the equivalent measures in the Control condition, these participants spent more time looking at the target of joint attention before attempting to establish eye contact. They also made more premature attempts at initiating joint attention before their virtual partner had returned his gaze to establish eye contact (cf. *Caruana, Brock & Woolgar, 2015*; *Caruana et al., 2017c*; *Caruana et al., 2017b*). As predicted, we found that both effects were reduced in the Computer group. These participants made fewer premature attempts and spent less time looking at the burglar before making eye contact. Importantly (and in contrast to the Responding condition), these effects were specific to the Social condition and could not, therefore, be explained in terms of an audience effect on performance. These findings are consistent with the view that participants in the Human group adopted an "intentional stance" towards the virtual character, and thus, expected their partner to be an intelligent and flexible agent who would follow their gaze cues, whether or not eye contact had been explicitly established.

Wykowska and colleagues (*2014*) have argued that when participants adopt an intentional stance towards an entity, this exerts a "top down" effect on the interaction, guiding the participant's predictions and expectations concerning the entity's behaviour. Thus, when individuals believe they are interacting with a human, they view their partner's behaviours as the product of an intentional and intelligent mind and engage in the mentalising processes that are normally recruited during human interactions. This in turn reinforces expectations about how the entity should behave. In the current context, this means that participants may have expected their partner to *know* that a prolonged dwell time on a particular location or rapid looking backward and forward between that location and the face indicated that they had found the burglar, even when eye contact was not explicitly established. In contrast, when interacting with a non-human entity, Wykowska and colleagues suggest that participants adopt a "design stance" in which they view the entity's behaviours as the product of an engineered system. Participants in the Computer group would not, therefore, have formed any expectations of their virtual partner, making them less likely to attempt initiating joint attention before eye contact had been established.

This interpretation of the eye-tracking data is also consistent with the subjective task ratings provided by participants at the completion of the experiment. Specifically, those in the Human group rated the Social condition task as being less intuitive than the Computer group. This makes sense, if we interpret the eye-tracking data as indicating a violation of the flexible responsive behaviour that the Human group participants expected from their partner.

The current findings compliment the recent neuroimaging studies of virtual joint attention interactions discussed earlier, which indicate that brain responses associated with the successful achievement of joint attention are moderated by beliefs about the human agency of the virtual partner (*Pfeiffer et al., 2014*; *Caruana, De Lissa & McArthur, 2017*). Our results are also broadly consistent with earlier studies that investigated the

neural correlates of mentalising by manipulating agency beliefs. For example, *Gallagher et al. (2002)* reported differential brain activity in the anterior paracingulate cortex during a computerised version of "stone, paper, scissors", depending on whether participants believed they were playing against a human or computer. Similar findings have been reported in other neuroimaging studies involving cooperative games (*McCabe et al., 2001*). However, the current results go further, indicating that human agency beliefs directly influence behaviour—how the participants interact with their virtual partner—and not simply how they evaluate the outcome of that interaction.

The results of the study indicate, therefore, that the ecological-validity of a virtual social interaction may depend on whether users believe their virtual partner represents another living human being. The design, development, and implementation of social simulations should therefore include consideration and, if necessary, evaluation of whether human agency beliefs facilitate or mitigate the achievement of the application's goals. The importance of these beliefs is likely to depend on the area of application. For instance, in social cognition and neuroscience research—as we establish directly in this paper—the adoption of human agency beliefs and an intentional stance appears to be an important ingredient when achieving an ecologically-valid measure of social cognition and behaviour (cf. *Caruana et al., 2017a*). Likewise, it would not be surprising that user behaviour be similarly affected in other social applications of virtual reality in the broader consumer space.

Currently, virtual reality applications are being developed to provide consumers with virtual teachers to automate education and training pipelines, virtual companions for the lonely, and virtual therapists for those without access to mental health care. It can be imagined in these applications, that the user's experience and the application's success would be influenced by whether they believe there is another human on the other side of the interaction providing genuine advice, friendship or support. Such beliefs could result in different degrees of value or trust placed in the utility of the training, companionship or therapy provided. Again, our subjective ratings provide some tentative supporting evidence, with participants in the Human group rating the task as being more pleasant, and their partner as more cooperative, than those in the Computer group.

It is also possible that, when a virtual interaction appears and feels sufficiently real, users may adopt an intentional stance, even when they know that their partner is not human. This is supported by previous studies of human–robot interaction which report an association between increased anthropomorphism and activation of brain regions implicated in mentalising processes (*Krach et al., 2008*). An interesting question for future research is whether the manipulation of agency beliefs has different effects depending on the type of virtual reality technology used and the degree of aesthetic and behavioural realism achieved (cf. *Georgescu et al., 2014*). For example, anthropomorphic stimuli are rated as more pleasant to look at, the more human-like they appear up to the point at which they behave almost, but not exactly like real humans. It is at this point that these stimuli can become aversive or subjectively unpleasant to look at—the so-called "uncanny valley effect". Furthermore, the tendency to anthropomorphise might be stronger for some users than others (*Waytz, Cacioppo & Epley, 2010*). Therefore, future work is required to

determine the conditions under which human agency beliefs impact on the virtual reality experience and how that may vary across individuals.

Virtual reality is a burgeoning industry that is promising many exciting applications for consumers, science and enterprise, particularly given its ability to realistically simulate social interactions between single users and virtual agents. In the current study, we investigate directly whether beliefs about a virtual partner's human agency can significantly influence the way in which users behave and feel—and present compelling evidence that at least in some interactive contexts, it does. Software developers and researchers attempting to simulate social interactions in virtual worlds need to be aware of the influence that these beliefs can have on user experience, and must evaluate how this might impact (positively or negatively) on the desired goal of the virtual reality application. Future research is needed to investigate how other factors such as social context, degree of immersion and avatar realism impact on user experience during virtual interactions.

### Funding
This work was supported by the Australian Research Council Centre of Excellence for Cognition and its Disorders [CE110001021]. The funders had no role in study design, data collection and analysis, decision to publish, or preparation of the manuscript.

### Grant Disclosures
The following grant information was disclosed by the authors:
Australian Research Council Centre of Excellence for Cognition and its Disorders: CE110001021.

### Competing Interests
Dr. Jon Brock is an Academic Editor for PeerJ.

### Author Contributions
- Nathan Caruana and Jon Brock conceived and designed the experiments, analyzed the data, contributed reagents/materials/analysis tools, wrote the paper, prepared figures and/or tables, reviewed drafts of the paper.
- Dean Spirou conceived and designed the experiments, performed the experiments, analyzed the data, wrote the paper, reviewed drafts of the paper.

### Human Ethics
The following information was supplied relating to ethical approvals (i.e., approving body and any reference numbers):

This study was approved by the Macquarie University Human Research Ethics Committee (Approval reference: 5201200021).

## Data Availability

Brock, Jon, and Nathan Caruana. 2017. "Human Agency Beliefs Influence Joint Attention Behaviour." Open Science Framework. https://osf.io/yqb7g/.

## Supplemental Information

Supplemental information for this article can be found online at http://dx.doi.org/10.7717/peerj.3819#supplemental-information.

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
