# Peer review of "Human agency beliefs influence behaviour during virtual social interactions"

_PeerJ, doi:10.7717/peerj.3819_

## Round 0.1 · original submission · Minor Revisions

Both reviewers commented on the quality of the study, analysis and writing. The authors should use their feedback to further improve their manuscript.

·

Basic reporting

Please see General comments for the author.

Experimental design

Please see General comments for the author.

Validity of the findings

Please see General comments for the author.

Additional comments

The manuscript focuses on the question how the interactive gaze behavior in a virtual environment (and the evaluation of the interaction) differs dependent on whether people believe that they are interacting with a human or with a computer. The study is well designed, descibed and analysed, it applies advanced techniques, and the use of scripted analysis supports the good scientific practice. The results are relevant for understanding the cognitive processes during social interactions and for applications of virtual realities in various contexts.

1. The research question is well defined, relevant and meaningful. While neural processing and evaluation of the virtual dynamic gaze interaction have been reported previously (Pfeiffer et al., 2014; Caruana, de Lissa & McArthur, 2017), the authors claim that there is a knowledge gap relating to the behavior during the virtual interaction.
However, from reading the Introduction (e.g., lines 164-169), it is not entirely clear to me how exactly this study differs from those, which were previously conducted by the authors and which used the same “burglar-task” (Caruana et al., 2015; Caruana, McArthur et al., 2017b; Caruana, Stieglitz Ham et al., 2017). The authors report that in their previous studies (please specify which ones also in line 164) the same results have been observed as reported in the current study (shorter dwell time in the Control condition, fewer premature saccades in the Control condition). In the Methods section (lines 210-212), the authors indicate that the experiment was identical to those used in these previous studies. If previous studies have demonstrated the same results with the same experiment, why these results need to be replicated in the current study? I guess that the previous studies did not include both the Human and the Computer condition, but only the Human condition, however this was not explicitly explained at this point (not until Discussion, line 404). For the readers who don’t know these previous studies in detail, this would be a helpful information.
2. In the Control condition, the participants were required to fixate the gray fixation point once completing their search for the burglar. The gray fixation point turned green after 500-1000ms (lines 265-266), which should be comparable to the establishing eye contact in the Social condition. However, for the Social condition, it is described that the virtual character was programmed to make 0-2 additional gaze shifts before establishing eye contact (after participants looked towards him; lines 231-232). Are these two delays really comparable? What was i) the minimal delay of “0 additional gaze” and ii) the maximal delay of “2 additional gazes” in the Social condition in ms? In other words, does it really take a minimum 500ms to immediately establish eye contact without additional gaze shifts, and does it really take not more than 1000ms to establish eye contact with 2 additional gaze shifts? This is an important detail because some of the main findings relate exactly to this event. For instance, shorter delays and smaller variance of establishing eye contact in IJAc condition could explain why participants had less premature saccades in IJAc than in IJA (for both Human, p < .005, and Computer, p = 046).
3. In any case, the authors should discuss why participants made more premature saccades in both the Computer and the Human group (disregarding the fact that this effect was greater in the Human condition; Human, p < .005, and Computer, p = 046). Importantly, the fact that more premature saccades were made in both Human and Computer groups does not necessarily align with the interpretation given in lines 440-443, because the Human group should not be expected to adopt the “intentional stance”.
4. Why were no tests for effects of factors on accuracy conducted (lines 333, 348)?
5. Why was the accuracy lower in the Control than in the Social condition only for RJA, but not for IJA? At least, this result supports authors’ interpretation in lines 416-418 and should be mentioned there. Because in the IJA condition, there was no need to recognize whether to follow a gaze cue or not, but only in the RJA condition, the lower accuracy in RJA-Social for both groups indicates that indeed some problems occurred in this process.
6. Related to the previous comment, is it possible that it was difficult to differentiate between eye contact (or direct gaze by the virtual character) and gaze towards any of the house locations? While the difference between direct gaze and gaze towards the left or right house may be easy to detect, the difference between direct gaze and gaze towards the upper or lower house in the middle may be more difficult. Should the authors not be sure about this, an inspection of the accuracy in trials with different gaze directions may be helpful to proof whether accuracy was reduced equally in all trials, or only in those trials with gaze towards the middle house. In addition, it may be helpful to include images of all possible gaze directions of the virtual character into the supplementary material (direct; left, middle and right for upper and lower).
7. The “uncanny valley” effect may be relevant for the discussion in lines 430-443, 460-463 and 498-500.
8. Do any of the results provide evidence for the tendency to anthropomorphise (line 502)? For instance, in lines 292-294, the authors write “The questions were designed to gauge the extent to which participants anthropomorphised the virtual character during their interaction with him”, however no results were discussed with respect to this question (the result referred to in Comment 3 above may be relevant for this question).

MINOR COMMENTS:

1. Methods, lines 194-195: Were subjects also screened for psychiatric disorders, or only for neurological ones?
2. Reporting exact p-values is more precise than reporting e.g. p < .005.
3. Fig.4: It would provide a better overview if those ratings were highlighted (e.g., with “*”), for which significant differences between the Human and the Computer group were detected.
4. “Overall task performance” may be too unspecified (line 396). Are the authors taking about accuracy here? Than this should be explicitly said because patters of eye-movements are indices of task performance as well (line 309-310).
5. Supplementary material 1: Including the card numbers into the figures would help to follow the descriptions relating to the cards (e.g., line 203).

·

Basic reporting

Basic Reporting:

The paper is very well and clearly written, the structure conforms to PeerJ standards.

Clear, unambiguous, professional English language is used thoughout, jargon is explained (e.g. "anthropomorphise", page 23, line 502), illustrative examples are provided (e.g. page 4, lines 71-73) to increase comprehensibility and emphasize the validity of the question.The authors have also meaningfully integrated examples from post-test questionnaire responses in order to contextualize results in the discussion.

Figures are relevant, high quality, well labelled and described but see coments to author for some suggestions for improvements.

Raw data are supplied on the Open Science FrameworkOSF with a link in the text.

Experimental design

Experimental Design:

The authors present a piece or original primary research and employ a cleverly designed and carefully controlled gaze contingent eye-tracking paradigm (previosuly validated), where participants believe they are interacting with either a virtual avatar of a real person or with a computer driven agent, to play a dynamic game of catching a burglar on the screen.

The research question is well framed and defined, relevant and meaningful. The authors investigate how beliefs about the interaction partner’s agency influence the actual behaviour (speed and gaze pattern during initiation and responding to joint attention) of participants during interactions, hence focusing on the production rather than the very often investigated perception side of social interaction. It is stated how the research fills an indetified knowledge gap, as relevant gaze-contingent paradigms are reviewed and open questions are formulated (e.g. page 5, line109/110).

Validity of the findings

Validity of the findings:

The study uses a novel yet validated paradigm and achieves interesting findings to advance research in the field, ultimately helping to better understand human social interaction. In particular, the research is useful for the recent approach emerging from social gaze research on studying the dual function of gaze.

The data is robust, statistically sound and controlled (for instance a non-social condition is included as well, supplementary material regarding instruction and cover story have been provided). Replications of findings are mentioned (e.g. page 19, line 403-404). Conclusions are well stated (e.g. page 19, line 400-402), linked to the original research question (e.g. page 19, 1st paragraph and mention of hypotheses when reporting results) and to previous research findings form the same lab and others (e.g. page 21-22, lines 467-476).

In the introduction section and the one for reporting of results I recommend some edits (see comments for the author).

Additional comments

General comments for the author:

In a manuscript titled "Human agency beliefs influence behaviour during virtual social interactions", the authors present a clearly designed and executed study that investigates several facets of how adult participants use social gaze cues to coordinate with a virtual other with either a human or artificial agency, in a novel joint attention task.

The authors find among other things that participants in the “human” condition are slower to make eye contact with their partner and more premature attempts at initiating joint attention compared to participants in the “Control” condition. The authors conclude that beliefs about the human agency of an interaction partner influences behaviour during social interactions.

The paper is very well and clearly written, the methodology and data analyses are correct.

Below I detail some comments:

A. Introduction/Methods:

1. The paradigm is described both in the methods section and in the introduction (pages 6/7), however, I feel the one in the introduction is too detailed and is confusing the comprehensibility of the description in the methods section. I recommend to shorten the description of the paradigm in the introduction and focus on describing it in the methods sections only.

2. In order to make an even stronger point on why using the gaze contingent paradigm AND FOR STUDYING BEHAVIOUR rather than mere perception the authors might want to consider research on the dual function of gaze (Gobel, Kim & Richardson, 2015) and claim that the results show that belief about agency affects the SIGNALLING function of gaze in interactions. The theory emphasizes the importance of investigating the signalling function of social gaze in order to get a deeper insight into complex social behaviours in more realistic situations. Due to methodological constraints so far, it has been difficult to emulate real-life social settings in the lab, and previous studies have primarily investigated the encoding function of social gaze (i.e. perception, how eyes capture or react to information from the environment). However there has been limited focus on understanding its signalling or communicative function (i.e. how eyes can send information back to other people), which is at the core of social interactions. Not just the current paradigm, but also the DVs are a step further in this direction, I would say.

B. Methods:

3. On page 9, can authors specify what the participant is told that Alan can see on the screen during the interaction in the “Human” condition? Did they think Alan can see an avatar of their face as well?

C. Results

4. On page 15, line 333, as well as page 16, line 348 the authors could complement the text under the subtitle for "accuracy” with a bit more information/stats, as they do for “saccadic reaction time”.

D. Discussion

5. On page 23, line 503, there must be research on this to cite. For instance, perhaps Waytz et al. 2014 https://www.ncbi.nlm.nih.gov/pmc/articles/PMC4021380/

6. On page 23, lines 500-502 consider citing perhaps this fMRI shoing ToM network in relation to increasing anthopopmorphism by Krach http://journals.plos.org/plosone/article?id=10.1371/journal.pone.0002597 and there is also work by Saygin on the action observation network https://academic.oup.com/scan/article/7/4/413/1738009/The-thing-that-should-not-be-predictive-coding-and

7. Perhaps the authors could discuss the difference in findings for responding and initiating.

E. Other Minor Comments

8. To my knowledge, if authors are cited within text, the citation should read (for example on page 5): “Pfeiffer and colleagues (2014)”, instead of “Pfeiffer et al., (2014)”

9. When reporting p-values for significant results, there is a mix of “p = “ and “p<”, I would advise to use p<.005/.001 etc for such significant results and p= for nonsignificant results example page 17, line 374. Also, P=< should be replaced with p<. I may be wrong and there may actually be a system to this, that I don't know of.

10. Typo in page 9, line 218, perhaps “doorS open”

11. On page 14 line 311, “accuracy” should be italicized

12. Captions for figures could be complemented. Figure 1 could include the information that this is a screenshot, explain that bottom row with blue doors is to be searched by participant, Results figures could include a specification if IJA and IJAc, and RJA and RJAc respectively. What are error bars?

13. Typo on page 21, line 464 "complement" instead of "compliment"

Note: On page 4, line108, the authors claim that the agency beliefs in the study by Pfeiffer et al., 2014 were confounded with task success (i.e. achieving joint attention). This was only the case in the naïve, not the cooperative condition. Nevertheless, the cooperative game in the present study is more similar to the naïve condition in that study, so the claim still holds.

---

## Round 0.2 · accepted · Accept

All comments have been adequately addressed.

·

Basic reporting

See General comments.

Experimental design

See General comments.

Validity of the findings

See General comments.

Additional comments

I would like to thank the authors for the effort put into replying to my comments. All my comments have been taken into account. For my opinion regarding review points 1 to 3 (Basic Reporting, Experimental Design, Validity of the Finding) please see my previous review.

·

Basic reporting

please see "general comments for the author"

Experimental design

please see "general comments for the author"

Validity of the findings

please see "general comments for the author"

Additional comments

The paper is very well and clearly written, the methodology and data analyses are correct. All my comments and concerns have been appropriately addressed and clarified in the rebuttal.